# P2 × 7 Receptor Inhibits Astroglial Autophagy via Regulating FAK- and PHLPP1/2-Mediated AKT-S473 Phosphorylation Following Kainic Acid-Induced Seizures

**DOI:** 10.3390/ijms21186476

**Published:** 2020-09-04

**Authors:** Duk-Shin Lee, Ji-Eun Kim

**Affiliations:** Department of Anatomy and Neurobiology, Institute of Epilepsy Research, College of Medicine, Hallym University, Chuncheon 24252, Korea; dslee84@hallym.ac.kr

**Keywords:** Bif-1, FAK inhibitor 14, LAMP1, p70S6K, PRAS40, Raptor, Rictor, siRNA

## Abstract

Recently, we have reported that blockade/deletion of P2X7 receptor (P2X7R), an ATP-gated ion channel, exacerbates heat shock protein 25 (HSP25)-mediated astroglial autophagy (clasmatodendrosis) following kainic acid (KA) injection. In P2X7R knockout (KO) mice, prolonged astroglial HSP25 induction exerts 5′ adenosine monophosphate-activated protein kinase/unc-51 like autophagy activating kinase 1-mediated autophagic pathway independent of mammalian target of rapamycin (mTOR) activity following KA injection. Sustained HSP25 expression also enhances AKT-serine (S) 473 phosphorylation leading to astroglial autophagy via glycogen synthase kinase-3β/bax interacting factor 1 signaling pathway. However, it is unanswered how P2X7R deletion induces AKT-S473 hyperphosphorylation during autophagic process in astrocytes. In the present study, we found that AKT-S473 phosphorylation was increased by enhancing activity of focal adhesion kinase (FAK), independent of mTOR complex (mTORC) 1 and 2 activities in isolated astrocytes of P2X7R knockout (KO) mice following KA injection. In addition, HSP25 overexpression in P2X7R KO mice acted as a chaperone of AKT, which retained AKT-S473 phosphorylation by inhibiting the pleckstrin homology domain and leucine-rich repeat protein phosphatase (PHLPP) 1- and 2-binding to AKT. Therefore, our findings suggest that P2X7R may be a fine-tuner of AKT-S473 activity during astroglial autophagy by regulating FAK phosphorylation and HSP25-mediated inhibition of PHLPP1/2-AKT binding following KA treatment.

## 1. Introduction

Astrocytes are the most abundant glial cells, which play vital roles in controlling extracellular ion/glutamate homeostasis, brain–blood barrier, energy metabolism, and synaptic function. Although astrocytes seem to be invulnerable to harmful stresses, astroglial damage is also reported in various neurological diseases, such as epilepsy, ischemia, and Alzheimer’s disease [1,2,3,4]. Indeed, Alzheimer has described irreversible astroglial degeneration showing vacuolization and disintegrated/beaded processes, and Cajal has termed it as “clasmatodendrosis” [5]. Recently, we have reported that clasmatodendrosis is lysosome-associated membrane protein 1 (LAMP1)-mediated astroglial autophagy [6]. Basically, autophagy is critical for energy homeostasis and cell survival, since it clears the damaged proteins and organelles and provides cellular energy and building blocks for biosynthesis by enabling the recycle of the degraded cellular components. However, excessive autophagy leads to cell death, which shows extensive cytoplasmic vacuolization and culminating with phagocytic uptake and consequent lysosomal degradation [7]. Clasmatodendritic astrocytes manifest eosinophilic cytoplasm with vacuolization and TUNEL-negativity [8]. Interestingly, clasmatodendrosis is exacerbated in P2X7 receptor (P2X7R, an ATP-gated ion channel) knockout (KO) mice and P2X7R antagonist-treated rats following kainic acid (KA) or pilocarpine injection [6,9]. Briefly, KA induces prolonged astroglial heat shock protein 25 (HSP25) expression due to impaired extracellular regulated kinase 1/2 (ERK1/2)-mediated specificity protein 1 (SP1) phosphorylation in P2X7R KO mice. Subsequently, upregulated HSP25 itself evokes endoplasmic reticulum stress and exerts 5′ adenosine monophosphate-activated protein kinase (AMPK)/unc-51 like autophagy activating kinase 1 (ULK1)-mediated autophagic pathways independent of mammalian target of rapamycin (mTOR) activity in astrocytes [6].

AKT is a negative regulator of autophagy that inhibits ULK1 activity [10]. AKT promotes glucose uptake and glycolysis, which stimulates ATP production and thereby inhibits AMPK activity [11]. Inhibition of AKT induces high level of autophagy in glioblastoma cells [12]. Thus, AKT and AMPK have opposing effects on the induction of autophagy. However, recent studies reveal that AKT exerts mTOR-independent autophagy [13]. In P2X7R KO mice, KA enhances AKT-serine (S) 473 phosphorylation, which leads to astroglial autophagy via glycogen synthase kinase-3β (GSK3β)/bax interacting factor 1 (Bif-1) signaling pathway [6]. Thus, it is noteworthy to unveil the role of P2X7R deletion-mediated AKT-S473 hyperphosphorylation in astroglial autophagy. In the canonical pathway, AKT is phosphorylated by phosphoinositide-dependent kinase-1 (PDK1), which requires production of phosphatidylinositol (3,4)-bisphosphate and phosphatidylinositol (3,4,5)-trisphosphate by phosphatidylinositol-4,5-bisphosphate 3-kinase (PI3K) activity, and dephosphorylated by phosphatase and tensin homolog deleted on chromosome 10 (PTEN) [14]. However, PIK3-independent AKT activation has been also reported [15]. In P2X7R KO mice, indeed, KA increases AKT-S473 phosphorylation in astroglial autophagic process, independent of PI3K and PTEN activities, which is abrogated by silencing HSP25 [6]. Similar to our previous study, other investigators have also reported PIK3-independent AKT activation [15]. Therefore, it is a key question left unanswered how P2X7R deletion and/or sustained HSP25 expression induce AKT-S473 hyperphosphorylation (activation) during autophagic process in astrocytes.

Here, we demonstrate for the first time that in P2X7R KO mice, KA increased AKT-S473 phosphorylation by enhancing activity of focal adhesion kinase (FAK), also known as protein tyrosine kinase 2 (PTK2), independent of mTOR complex (mTORC) 1 and 2 activities. In addition, HSP25 overexpression in P2X7R KO mice acted as a chaperone of AKT, which sustained AKT-S473 phosphorylation by inhibiting the pleckstrin homology domain and leucine-rich repeat protein phosphatase (PHLPP) 1- and 2-binding to AKT following KA injection. Therefore, we suggest the novel signaling pathway for P2X7R deletion-mediated AKT-S473 hyperphosphorylation, which may facilitate astroglial autophagy following KA treatment.

## 2. Results

### 2.1. P2X7R Deletion Facilitates Astroglial Autophagy Following KA Injection

The degree of seizure activity affects astroglial responses including HSP25 induction [16]. Thus, we evaluated the seizure susceptibilities of wild-type (WT) and P2X7R KO mice in responses to KA. Consistent with our previous report [6], there was no difference in the latency of seizure onset and the normalized total power of electroencephalogram (EEG) during seizures in response to KA between WT and P2X7R KO mice (Figure 1A–C).

To assess the purity of isolated astrocytes, we performed immunohistochemistry using glial fibrillary acidic protein (GFAP, an astroglial marker), neuronal nuclear antigen (NeuN, a neuronal marker), ionized calcium binding protein-1 (Iba-1, a microglial marker), and RIP (an oligodendroglial marker) antibodies. The isolated cells mainly showed GFAP signals, but rarely exhibited NeuN, Iba-1, or RIP positivity (Figure 1D). Western blot study demonstrated the upregulations of HSP25, phospho (p)-HSP25, and LAMP1 expression in isolated astrocytes obtained from P2X7R KO mice without affecting p-HSP25/HSP25 ratio after KA injection, as compared to WT mice (*p* < 0.05, Student *t*-test; *n* = 7; Figure 1E,F and Appendix A). Immunostaining also revealed that KA increased astroglial HSP25 expression in KO mice more than WT mice, although its expression was rarely observed in both animals under physiological condition (*p* < 0.05, one-way ANOVA; *n* = 7; Figure 1G,I). KA also elevated astroglial LAMP1 expressions in KO mice following KA injection (*p* < 0.05, one-way ANOVA; *n* = 7; Figure 1H,J). Since LAMP1 is important for the autophagolysosomal pathway [17], these findings indicate that KA-induced upregulation of HSP25 expression may exert astroglial autophagy in P2X7R KO mice, independent of seizure activity.

Because of the insufficient amount of isolated astrocytes obtained from the bilateral hippocampi for Western blot, we used the bilateral cerebral cortices of an individual mouse to dissociate astrocytes. Thus, we addressed the possibility of the distinct astroglial responses between the cortical and the hippocampal astrocytes in P2X7R KO mice. Consistent with our previous study [6] and the immunohistochemical data in the present study (Figure 1G–J), the Western blot data obtained from the whole hippocampus of P2X7R KO mice demonstrated the upregulations of HSP25 and LAMP1 expressions following KA injection. The alterations in HSP25 and LAMP1 expression in the whole hippocampus were compatible with those in isolated astrocytes in P2X7R KO mice (*p* < 0.05, one-way ANOVA; *n* = 7; Figure 2A,B and Appendix A). These findings indicate that the data obtained from isolated astrocytes may represent the responses of hippocampal astrocytes.

To confirm the role of P2X7R knockout in HSP25 and LAMP1 expressions, we applied A740003 (a P2X7R antagonists) to WT mice. A740003 increased HSP25 and LAMP1 expressions in the whole hippocampus of WT mice following KA injection (*p*  <  0.05 vs. vehicle, one-way ANOVA; Figure 3A,B and Appendix A). These findings suggest that blockade/deletion of P2X7R may accelerate HSP25 induction and astroglial autophagy.

### 2.2. P2X7R Deletion Enhances FAK-Mediated AKT-S473 Phosphorylation in Astrocytes Independent of Pi3k/Pdk1 Signaling Pathway

Next, we investigated if P2X7R deletion would influence PDK1 activity that phosphorylates AKT [14], although no differences in PI3K and PTEN activities have been observed in the hippocampi between WT and P2X7R KO mice following KA injection [6]. Under physiological condition, p-PI3K-tyrosine (Y) 458 and p-PDK1-S241 levels in WT astrocytes are similar to those in KO astrocytes (Figure 4A,B and Appendix A). KA injection reduced PI3K-Y458, but not PDK1-S241, phosphorylation in both astrocytes (*p* < 0.05 vs. control animals, one-way ANOVA; *n* = 7; Figure 4A,B and Appendix A). However, KA significantly increased AKT-S473 and GSK3β-S9 phosphorylations and Bif-1 expression in KO, but not WT, astrocytes (*p* < 0.05 vs. WT astrocytes, one-way ANOVA; *n* = 7; Figure 4A,B and Appendix A). Consistent with our previous study [6], in the whole hippocampus of P2X7R mice, KA increased AKT-S473 and GSK3β-S9 phosphorylations and Bif-1 expression (*p* < 0.05 vs. control mice, one-way ANOVA; *n* = 7; Figure 2A,B and Appendix A). A740003 also elevated AKT-S473 and GSK3β-S9 phosphorylations and Bif-1 expression in the whole hippocampus of WT mice following KA injection (*p* < 0.05 vs. vehicle, one-way ANOVA; *n* = 7; Figure 3A,B and Appendix A). These findings indicate that deletion/blockade of P2X7R may activate AKT after KA injection, independent of PI3K/PDK1 signaling pathway.

On the other hand, FAK is one of the kinases for AKT, which is required for autophagy [18]. FAK activation is required for the phosphorylations on several tyrosine (Y) residues in kinase domain, including Y397, 576, and 577. Among them, Y397 is autophosphorylation site, and other sites are phosphorylated by Src [19]. Since P2X7R antagonists increase FAK activity [20], we validated whether P2X7R affects astroglial FAK activity (phosphorylation) following KA injection. Under physiological condition, FAK expression and its phosphorylation were similarly observed between WT and P2X7R KO astrocytes (Figure 4A,B and Appendix A). Unlike WT astrocytes, KA increased FAK-Y397 and -Y576 phosphorylations in KO astrocytes. KA-induced FAK-Y397 phosphorylation was more prominent than Y576 phosphorylation (*p* < 0.05 vs. WT astrocytes, one-way ANOVA; *n* = 7; Figure 4A,B and Appendix A). In the whole hippocampus of P2X7R mice, KA increased FAK-Y397 phosphorylation (*p* < 0.05 vs. control mice, one-way ANOVA; *n* = 7; Figure 2A,B and Appendix A). In addition, A740003 elevated it in the whole hippocampus of WT mice following KA injection (*p* < 0.05 vs. vehicle, one-way ANOVA; *n* = 7; Figure 3A,B and Appendix A). Immunostaining also revealed that KA upregulated astroglial FAK-Y397 phosphorylation in the hippocampus of KO mice (Figure 4C). Thus, it is likely that P2X7R deletion may increase AKT-S473 phosphorylation by aberrant FAK activation. To confirm this, we applied FAK inhibitor 14 (FI14) in KO mice following KA injection. FI14 decreased AKT-S473 and GSK3β-S9 phosphorylations in KO astrocytes after KA injection (*p* < 0.05 vs. vehicle, one-way ANOVA; *n* = 7; Figure 5A,B and Appendix A). In addition, FI14 diminished Bif-1 and LAMP1 expressions in KO astrocytes following KA injection (*p* < 0.05 vs. vehicle, one-way ANOVA; *n* = 7; Figure 5A,B and Appendix A), while it could not affect HSP25 expression and p-HSP25 level (Figure 5A–C and Appendix A). Thus, our findings suggest that P2X7R deletion may promote FAK-mediated AKT phosphorylation during astroglial autophagy induced by KA, although FAK activation was not involved in prolonged HSP25 expression and its phosphorylation.

### 2.3. P2X7R Does Not Affect mTORC-Mediated AKT Phosphorylation Following KA Injection

Together with AKT-S473 phosphorylation, KA also facilitates astroglial autophagy via AMPK activation in P2X7R KO mice [6]. AMPK activity influences AKT phosphorylation via mTORC1/p70S6 kinase (p70S6K)/mTORC2 signaling pathway. Briefly, AMPK inhibits mTORC1 through the phosphorylation of regulatory associated protein of mTOR (Raptor; a mTORC1 partner protein) at S792 [21], which in turn abrogates p70S6K phosphorylation [22]. This signaling cascade enhances AKT-S473 phosphorylation by reducing rapamycin-insensitive companion of mammalian target of rapamycin (Rictor; a mTORC2 partner protein)-T (threonine) 1135 phosphorylation, since p70S6K directly phosphorylates Rictor at T1135 site [23]. Thus, it is likely that P2X7R deletion would upregulate AKT-S473 phosphorylation by affecting mTORC1/2-mediated pathway. In the present study, AMPK and ULK1 expressions and their phosphorylations were similarly observed in WT- and P2X7R KO astrocytes under physiological condition (Figure 6A,B and Appendix A). KA increased AMPK-T172 and ULK1-S555 phosphorylations in P2X7R KO astrocytes without changing their expression levels (*p* < 0.05 vs. vehicle one-way ANOVA; *n* = 7; Figure 6A,B and Appendix A), while it did not affect AMPK and ULK1 expressions/phosphorylations in WT astrocytes (Figure 6A,B and Appendix A). Unlike AMPK, P2X7R deletion did not alter phosphorylations/expressions of Raptor, Rictor, and p70S6K in astrocytes under physiological and post-KA conditions (Figure 6A,B and Appendix A). In addition, KA did not affect phosphorylations of mTOR and proline-rich AKT substrate of 40 kDa (PRAS40; known as an endogenous inhibitor for mTORC1) [24] in WT and P2X7R KO astrocytes (Figure 6A,B and Appendix A). These findings indicate that both AMPK and mTORC1/p70S6K/mTORC2 signaling pathway may not be involved in AKT-S473 hyperphosphorylation in P2X7R KO astrocytes following KA injection.

### 2.4. P2X7R Deletion Cannot Influence PHLPP1/2 Expression in Astrocytes Following KA Injection

PHLPP1/2 are ubiquitous serine/threonine phosphatases that dephosphorylate AKT at S473 site [25]. In our previous study [6], P2X7R deletion promotes astroglial autophagy by prolonged HSP25 transactivation due to abrogation of ERK1/2-mediated SP1-T739 phosphorylation. This is because HSP25 expression relies on SP1 binding to its promoter region, and SP1-T739 phosphorylation reduces the SP1 DNA-binding ability [26,27]. Interestingly, SP1 also increases PHLPP promoter activity and its protein level, and facilitates AKT dephosphorylation [28]. Therefore, it is likely that P2X7R would regulate AKT-S473 phosphorylation by inhibiting ERK1/2-SP1-mediated PHLPP1/2 transactivations following KA injection. In the present study, ERK1/2 phosphorylation level was reduced in P2X7R KO astrocytes, but not WT astrocytes, after KA treatment (*p* < 0.05 vs. control animals one-way ANOVA; *n* = 7; Figure 7A,B and Appendix A). Similar to ERK1/2 phosphorylation, KA decreases SP1-T739 phosphorylation in the P2X7R KO astrocytes more than in WT astrocytes (*p* < 0.05 vs. control mice and KA-treated WT mice, respectively, one-way ANOVA; *n* = 7; Figure 7A,B and Appendix A). However, KA injection did not affect expression levels of PHLPP1 and PHLPP2 in both WT and KO astrocytes (Figure 7A,B and Appendix A). These findings suggest that P2X7R deletion may not affect ERK1/2-SP1-mediated PHLPP1/2 transactivations, although P2X7R knockout facilitates ERK1/2-SP1-mediated HSP25 overexpression.

### 2.5. HSP25 Binding to AKT Abrogates AKT-S473 Dephosphorylation

The binding of HSP25 regulates AKT protein kinase activity [29], and p-HSP25 forms a new complex with AKT securing AKT’s natural conformation and enzymatic activity [30]. Thus, HSP25 is believed to affect AKT-S473 phosphorylation [31]. Indeed, HSP25 siRNA attenuates KA-induced AKT-S473 phosphorylation in P2X7R KO mice [6]. Furthermore, HSP25 expression and its phosphorylation affect FAK activity [32]. Therefore, we investigated the remaining question how HSP25 regulates AKT-473 phosphorylation and whether it affects FAK phosphorylation during astroglial autophagy in P2X7R KO mice.

Consistent with our previous study [6], HSP25 siRNA effectively reduced HSP25 expression and HSP25 phosphorylation in P2X7R KO astrocytes following KA injection (*p* < 0.05 vs. control siRNA; *n* = 7; Figure 8A–C and Appendix A). Furthermore, HSP25 knockdown abolished AKT-S473, and GSK3β-S9 phosphorylations without changing their expression level (*p* < 0.05 vs. control siRNA, one-way ANOVA; *n* = 7; Figure 8A,B and Appendix A). HSP25 siRNA also diminished Bif-1 and LAMP1 expressions following KA injection (*p* < 0.05 vs. control siRNA, one-way ANOVA; *n* = 7; Figure 8A,B and Appendix A). However, HSP25 knockdown did not affect FAK expression and phosphorylations (Figure 8A,B and Appendix A). Coimmunoprecipitation data revealed that HSP25 bound with AKT, but not FAK, which was abrogated by FI14 (*p* < 0.05 vs. vehicle, Student *t*-test; *n* = 7; Figure 9A,B and Appendix A). Furthermore, HSP25 siRNA increased the binding of AKT with PHLPP1/2 (*p* < 0.05 vs. control siRNA, Student *t*-test; *n* = 7; Figure 9A,B and Appendix A). Taken together, these findings suggest that HSP25 may function as a chaperon to prevent AKT-S473 dephosphorylation by PHLPP1/2, rather than an indispensable factor for AKT phosphorylation.

## 3. Discussion

The major findings in the present study are P2X7R deletion facilitated astroglial autophagy via FAK- and HSP25-mediated regulations of AKT-S473 phosphorylation following KA injection, independent of PI3K/PDK1- and mTORC1/2 signaling pathways (Figure 10).

In the canonical pathway, AKT activation requires PI3K-dependent phosphorylation of threonine (T) 308 and S473 sites. When phosphatidylinositols are produced by a PI3K, PDK1 is recruited to the plasma membrane and leads to AKT-T308 phosphorylation [33]. The AKT-S473 phosphorylation is regulated by PDK1 or mTORC2, once known as the elusive PDK2 [33]. Consistent with our previous study [6], the present study demonstrates that KA specifically increased astroglial AKT-S473 phosphorylation in P2X7R KO astrocytes. However, KA did not affect PDK1-S241 phosphorylation in both WT and P2X7R KO astrocytes, although it reduced PI3K activity (phosphorylation) in both groups. Since PDK1-S241 site is constitutively phosphorylated by an autophosphorylation reaction in *trans* and PDK1 does not always require phosphatidylinositols for its activities [34,35], our findings indicate that the PI3K/PDK1 pathway may not be involved in KA-induced AKT-S473 hyperphosphorylation in P2X7R KO astrocytes. However, mitogen-activated protein kinase 2 (MK2) and a complex with the integrin-linked kinase (ILK) and interact with protein kinase C-ζ (PKC-ζ) can phosphorylate AKT on S473 site independent of PI3K/PDK1 [31,36,37]. Thus, it is not excluded the possibility that these noncanonical signaling pathways would be involved in AKT-S473 hyperphosphorylation in P2X7R KO astrocytes. It will be interesting to see if these PI3K/PDK1-independent pathways impact AKT phosphorylation in P2X7R KO astrocytes.

In the present study, P2X7R KO astrocytes showed the increases in GSK3β phosphorylation and expressions of Bif-1 and LAMP1, accompanied by AKT-S473 hyperphosphorylation after KA treatment. Since AKT-mediated GSK3β phosphorylation (reducing its activity) transactivates Bif-1 that modulates autophagy by regulating autophagosome formation [13,38,39], our findings indicate that AKT-S473 hyperphosphorylation may promote GSK3β/Bif-1-mediated autophagic pathway in P2X7R KO astrocytes after KA injection. On the other hand, FAK is the upstream regulatory molecules for PI3K/PDK1-mediated AKT activation [40]. Autophosphorylation of FAK at Y397 site creates a potential binding site for the SH2 domains of the p85 subunit of PI3K [41]. Phosphorylation of the p85 subunit of PI3K by FAK activates the p110 catalytic subunit of PI3K [42], and subsequently phosphorylates PDK1 and AKT, especially at T308 site [43]. Interestingly, P2X7R inhibition increases FAK activity by the autophosphorylation of FAK at Y397 site [20]. FAK is also responsible for AKT-S473 phosphorylation via PI3K/PDK1-independent signaling pathway [43,44]. In the present study, KA upregulated FAK-Y397 and -Y576 phosphorylations in P2X7R KO astrocytes. KA elevated Y397 phosphorylation more than Y576 phosphorylation. Furthermore, FI14 inhibited autophagy in P2X7R KO astrocytes with diminishing phosphorylations of AKT-S473 and GSK3β-S9 and expressions of Bif-1 and LAMP1 following KA injection. Since FAK directly binds to AKT and phosphorylates S473 site independent of PI3K/PDK1 activity [43,44], our findings indicate that P2X7R deletion may increase FAK autophosphorylation in response to KA, which is required for AKT-mediated astroglial autophagy, independent of PI3K/PDK1 signaling pathway.

The mTOR is an essential regulator of many major cellular functions, such as metabolism, growth, proliferation, and autophagy. In executing this role, mTOR participates in two distinct complexes: mTORC1 and mTORC2 [22,45]. Both mTORCs contain the catalytic mTOR subunit, mammalian lethal with SEC13 protein 8 (MLST8, also known as G protein beta subunit-like (GβL)), DEP domain-containing mTOR-interacting protein (DEPTOR), and the TELO2 interacting protein 1 (Tti1)/Telomere maintenance 2 (Tel2) complex. Raptor and PRAS40 are specific to mTORC1, while Rictor, mammalian stress-activated protein kinase interacting protein 1 (SIN1) and protein observed with Rictor (Protor) 1/2 are specific to mTORC2 [45]. AKT directly phosphorylates mTOR on S2448, which is frequently used as a marker of mTORC1 and mTORC2 activation [14,22]. However, mTORC1 and mTORC2 activities are also regulated by Raptor and Rictor phosphorylations. Raptor-S792 phosphorylation by AMPK inhibits mTORC1 activity independent of mTOR-S2448 phosphorylation [21]. This deactivation of mTORC1 inhibits p70S6K [22], which abrogates Rictor-T1135 phosphorylation, and in turn activates mTORC2-mediated AKT-S473 phosphorylation [23]. In the present study, KA increased AMPK activity (T172 phosphorylation) in P2X7R KO astrocytes. However, KA did not affect phosphorylations of Raptor, Rictor, PRAS40, p70S6K, and mTOR in P2X7R KO astrocytes. Thus, our findings suggest that P2X7R deletion may facilitate astroglial autophagy independent of mTORC1/2 pathways, and that mTORC2 may not participate in AKT-S473 hyperphosphorylation in P2X7R KO astrocytes following KA injection. However, Rictor can form a complex with the ILK and interact with PKC-ζ, which regulate AKT-S473 phosphorylation in an mTORC2-independent fashion [36,37]. Thus, it is presumable that ILK- and/or PKC-ζ-mediated signaling pathway may be involved in AKT hyperactivation in P2X7R KO astrocytes following KA injection. In addition, FAK functionally suppresses mTORC2, but not mTORC1 [46]. FAK is also required for optimal signaling to the PDK1/AKT/p70S6K pathway independent of mTOR during autophagy process [47]. In the present study, we found that KA could not affect mTOR, Raptor and Rictor phosphorylations, but increased FAK activity in P2X7R KO astrocytes. Thus, our findings also suggest that increased FAK activity in P2X7R KO astrocytes may lead to KA-induced AKT-S473 hyperphosphorylation, independent of mTORC1 and 2 activities.

PHLPP1/2 are ubiquitous serine/threonine phosphatases, which inhibit AKT activity by S473 dephosphorylation [25]. Interestingly, P2X7R deletion abrogates ERK1/2-mediated SP1-T739 phosphorylation (inactivation) [6], which decreases PHLPP promoter activity and its protein level and subsequently leads to AKT hyperphosphorylation [28]. In the present study, KA reduced ERK1/2 and SP1 phosphorylations in P2X7R KO astrocytes. However, P2X7R deletion did not affect astroglial PHLPP1/2 expression levels under physiological condition and after KA injection. Thus, these findings indicate that P2X7R deletion may not affect astroglial ERK1/2-SP1-mediated PHLPP1/2 transactivations. On the other hand, HSP25 functions as a molecular chaperone [26] and forms complexes with AKT [30]. In addition, p-HSP25 is responsible for a new complex with AKT securing AKT’s natural conformation and enzymatic activity [30]. Indeed, the amounts of HSP25-AKT binding correlates with AKT protein kinase activity [29], and HSP25 siRNA inhibits AKT phosphorylation [6,48]. However, HSP25-AKT interaction is not required to promote AKT activation [49], although HSP25 regulates AKT-S473 phosphorylation [31]. Furthermore, HSP25 does not interact with PI3K at base line or after metabolic stress even in cells with increased HSP25 expression [49]. In the present study, P2X7R deletion and A740003 resulted in prolonged astroglial HSP25 expression following KA injection, which was concomitant with increased HSP25 phosphorylation. Consistent with our previous study [6], silencing HSP25 abolished AKT-S473 and GSK3β-S9 phosphorylations, and diminished Bif-1 and LAMP1 expressions in P2X7R KO mice following KA injection. Furthermore, HSP25 siRNA increased AKT-PHLPP1/2 bindings without altering their expression levels. Considering that HSP25 acts as a scaffolding protein to AKT and MK2 in a manner that requires HSP25-AKT interaction [31], our findings suggest that HSP25 may play a role as a chaperon to abolish AKT-PHLPP1/2 bindings rather than an indispensable factor for AKT phosphorylation, which maintains AKT-S473 phosphorylation in biologically active conformation by inhibiting PHLPP1/2-mediated dephosphorylation.

It has been also reported that HSP25 overexpression affects FAK phosphorylation [50]. However, the role of HSP25 in FAK phosphorylation has been still controversial. A phosphomimetic form of HSP25 increases FAK phosphorylation [50] or abolishes it [51]. In addition, wild-type HSP25 and nonphosphorylatable HSP25 does not affect FAK phosphorylation [32,50]. In the present study, we found that HSP25 did not bind to FAK and that HSP25 siRNA did not affect FAK-Y397 and -Y576 phosphorylations in P2X7R KO astrocytes after KA injection. However, FI14 reduced the HSP25-AKT binding, accompanied by diminished AKT-S473 phosphorylation. Thus, it is likely that prolonged HSP25 expression may not affect FAK phosphorylation, but may preserve FAK-mediated AKT-S473 phosphorylation in P2X7R KO astrocytes.

Murine/rodent HSP25 is a homologue of human HSP27. HSP25 induction is an indicative of the early astroglial responses including energy-consuming protein synthesis and is one of the highly sensitive and reliable molecules of full development of status epilepticus (SE). Since HSP25 is the main chaperone in astrocytes and plays a role in blockade of mitochondrial damage and apoptosis in naïve astrocytes, HSP25 induction provides insight into the astroglial invulnerability in response to harmful insults [16,52,53]. Indeed, HSP27 overexpression is a negative prognostic marker of astroglioma, which is regarded as a resistant mechanism for astroglioma cells against antitumor agents by activating autophagy [54,55]. In addition, HSP27 activates the autophagy-lysosomal pathway in abnormal astrocytes containing Rosenthal fibers in patients of Alexander disease that is an untreatable and fatal neurodegenerative disorder showing seizures caused by heterozygous mutations of GFAP gene [56,57,58]. Although autophagy is a catabolic and survival process for bulk degradation of aberrant organelles and protein aggregates, excessive or unquenched autophagy results in nonapoptotic programmed cell death (type II programmed cell death) independent of caspase activity in various cells including cancer cells [59,60,61]. Indeed, impaired clearance of HSP25 reduces cell viability in astrocytes [62]. Thus, the proper regulations of HSP25/27 expression and its related astroglial autophagy may be one of potential therapeutic strategies for treatment and/or mitigation of symptoms in patients with Alexander disease, astroglioma, and other neurodegenerative disorders with astrocyte involvement.

The limitations of the present study are the possibility that the dissociation procedure would affect signaling pathway in astrocytes and that KA would result in the loss of ATPase Na^+^/K^+^ transporting subunit beta 2 (ATP1B2) on the cell body and processes of astrocytes recognized by astrocyte specific cell surface antigen 2 (ACSA-2) antibody [63]. In the present study and our previous study [6], the data obtained from the whole hippocampus were compatible with those from isolated astrocytes. Furthermore, the isolation method using ACSA-2 antibody is regarded as less damaging and more effective with brain tissue and is highly efficient, yielding 300,000 ultrapure astrocytes per mouse cortex dissociated, with an average viability of 86% and purity of >95%. In addition, ACSA-2 signal remains constant in the sample obtained from models of a stab wound injury, Alzheimer’s disease, spinal cord injury, stroke, and lipopolysaccharide-induced inflammation [63]. Thus, it is likely that immunoaffinity-based method using ACSA-2 antibody may dissociate astrocyte with less affecting signaling pathways and yields of isolation following KA treatment.

## 4. Materials and Methods

### 4.1. Experimental Animals and Chemicals

Male C57BL/6J (WT) and P2X7R KO mice (60–90-day-old, 25–30 g, The Jackson Laboratory, USA) were used in the present study. Mice were given a diet and water ad libitum under controlled conditions (22 ± 2 °C, 55% ± 5% humidity, and 12-h light/12-h dark cycle). Animal protocols were approved by the Institutional Animal Care and Use Committee of Hallym University (Chuncheon, South Korea, Code number: Hallym 2018-3, 30th April 2018). All reagents were obtained from Sigma-Aldrich unless otherwise indicated. Appendix A is a list of the primary antibodies used in the present study.

### 4.2. Seizure Induction and Infusions of Drug and siRNA Oligonucleotide

Animals were injected a single dose of KA (25 mg/kg, i.p.) [6]. As controls, mice were treated with saline instead of KA [6,64]. Two hours after KA injection, animals received diazepam (10 mg/kg, i.p.) to terminate seizures. Two days after KA injection, mice were anesthetized with isoflurane (1–2% in O_2_ and N_2_O) and placed in a stereotaxic frame. Each animal was implanted with a Brain Infusion Kit 3 (Alzet, Cupertino, CA, USA) inserted into the lateral cerebral ventricle 1.0 mm lateral to the bregma and connected to an Alzet osmotic minipump (model 1007D, Cupertino, CA, USA) containing each compound or siRNA; vehicle, FI14 (a FAK inhibitor, 1 mM), A740003 (a P2RX7 antagonist, 10 μM, Santa Cruz Biotechnology Inc., Dallas, TX, USA), mouse *HSP25* siRNA (Sense, 5′- AAUAAAAGUUGCAAGCUACUU-3′; antisense, 5′-GUAGCUUGCAACUUUUAUUUU-3′) or a non-silencing (control) RNA. The doses of drugs and siRNA were chosen based on our previous and preliminary studies indicating that administration of up to the chosen dose was well tolerated and no signs of neurotoxicity (hind-limb paralysis, vocalization, food intake, or neuroanatomical damage) were observed. Seven-day after KA injection, animals were used for astroglial isolation, Western blot, coimmunoprecipitation, and immunohistochemistry [6,64].

### 4.3. Electrophysiology

Under isoflurane anesthesia (1–2% in O_2_ and N_2_O), animals were stereotaxically implanted a monopolar electrode into the left dorsal hippocampus (i.e., 2.0 mm posterior, 1.5 mm lateral, 2.0 mm depth). Three days after surgery, mice were given a single dose of KA aforementioned after establishing a stable baseline for at least 30 min, and latency or seizure onset and total power were measured from each animal. EEG signals were recorded with a DAM 80 differential amplifier (0.1–1000 Hz bandpass; World Precision Instruments, Sarasota, FL, USA), and the data were digitized (1000 Hz) and analyzed using LabChart Pro v7 software (ADInstruments, Bella Vista, New South Wales, Australia). Latency of seizure onset was defined as the time point showing more than 3 s and consisting of a rhythmic discharge between 4 and 10 Hz with amplitude of at least two times higher than the baseline EEG. Total EEG power was normalized by the baseline power obtained from each animal. Spectrograms were automatically calculated using a Hanning sliding window with 50% overlap by LabChart Pro v7 [6,64].

### 4.4. Astroglial Isolation

In a previous study [6], we had found that the amount of isolated astrocytes obtained from the bilateral hippocampi of a mouse was not enough for Western blot. Thus, we used the bilateral cerebral cortices of an individual mouse to purify astrocytes. Mice were anesthetized and decapitated. Bilateral cerebral cortices were removed and dissociated with adult brain dissociation kit (Miltenyi Biotec, Bergisch Gladbach, Germany, #130-107-677), in turn astrocytes were isolated with anti-ACSA-2 kit (Miltenyi Biotec, Bergisch Gladbach, Germany, #130-097-678), according to the manufacturer’s guidelines. Briefly, up to 10^5^ dissociated cells were suspended in 150 μL 0.5% BSA in PBS buffer and incubated with ACSA-2 microbeads for 25 min at 4 °C. Then, cells were applied to a MS column fitted in magnetic cell separator. Following column removal from the magnetic separator, astrocytes were eluted in 1.5 mL buffer. To evaluate the purity of isolated astrocytes, 50 μL of eluted samples were smeared on the slides, fixed by 4% paraformaldehyde in phosphate buffer in 30 min, and stained using GFAP, NeuN, Iba-1, or RIP antiserum (see Section 4.6).

### 4.5. Coimmunoprecipitation and Western Blot

Since the amount of isolated astrocytes was insufficient, we used the whole hippocampus to perform coimmunoprecipitation. The hippocampal tissues were lysed in radioimmunoprecipitation assay buffer (50 mM Tris–HCl pH 8.0; 1% Nonidet P-40; 0.5% deoxycholate; 0.1% SDS, Thermo Fisher Scientific, USA) containing protease inhibitor cocktail (Roche Applied Sciences, Branford, CT, USA), phosphatase inhibitor cocktail (PhosSTOP^®^, Roche Applied Science, Branford, CT, USA), and 1 mM sodium orthovanadate. Protein concentrations were calibrated using a Micro BCA Protein Assay Kit (Pierce Chemical, Rockford, IL, USA), and equal amounts of total proteins were incubated with HSP25 or AKT antibody (Appendix A) and protein G Sepharose beads overnight at 4 °C. Beads were collected by centrifugation, eluted in 2× SDS sample buffer, and boiled at 95 °C for 5 min. Thereafter, Western blot was performed as below. Isolated astrocytes were homogenized for Western blot by the same method. To address the possibility of the distinct astroglial responses between the cortical and the hippocampal astrocytes and the role of P2 × 7R deletion in astroglial autophagy, the whole hippocampi of P2 × 7R KO mice, vehicle- and A740003-treated WT mice were also used. Western blot was performed by the standard protocol. Briefly, aliquots were loaded into a polyacrylamide gel. After electrophoresis, gels were transferred to nitrocellulose transfer membranes. Membranes were incubated with primary antibody. Thereafter, membranes were reacted with an HRP-conjugated secondary antibody and ECL kit (GE Healthcare, Piscataway, NJ, USA). The bands were detected and quantified on ImageQuant LAS4000 system (GE Healthcare, Piscataway, NJ, USA). The rabbit anti-β-actin primary antibody (1:5000) was used as internal reference. Intensity measurements were represented as the mean gray-scale value and normalized against β-actin [6,64].

### 4.6. Double Immunofluorescence Study

Control and KA-treated animals were perfused via the ascending aorta with 200 mL 4% paraformaldehyde in phosphate buffer. The brains were removed and cryoprotected by infiltration with 30% sucrose overnight. Thereafter, the tissues were sectioned with a cryostat at 30 μm and consecutive sections were collected in six-well plates containing PBS. Sections were incubated in a mixture of antisera (glial fibrillary acidic protein (GFAP) + HSP25, GFAP + LAMP1, or GFAP + p-FAK-Y397) in PBS containing 0.3% Triton X-100 overnight at room temperature. After washing, sections were incubated in a mixture of FITC- and Cy3-conjugated IgG (or streptavidin, Jackson ImmunoResearch Laboratories Inc., West Grove, PA, USA; diluted 1:250) for 2 h at room temperature. To establish the specificity of the immunostaining, a negative control test was carried out with preimmune serum instead of the primary antibody. No immunoreactivity was observed for the negative control in any structures. All experimental procedures in this study were performed under the same condition and in parallel. Images were captured using an Axio Image M2 microscope (Carl Zeiss Korea, Seoul, South Korea). Manipulation of the images was restricted to threshold and brightness adjustments to the whole image [6].

### 4.7. Statistics

After evaluating the values on normality using Shapiro–Wilk *W*-test, Student’s *t*-test (comparisons of latency of seizure onset and Western blot data), repeated measured ANOVA (comparisons of total EEG power), or one-way ANOVA (comparisons of Western blots) was used to analyze statistical significance. Bonferroni’s test was used for post hoc comparisons. A *p*-value less than 0.05 was considered to be significant [6,64].

## 5. Conclusions

The present study demonstrates that P2 × 7R deletion-mediated AKT-S473 hyperphosphorylation, which facilitates astroglial autophagy, may be a consequence from FAK activation and a chaperone activity of overexpressing HSP25 to prevent AKT-PHLPP1/2 binding, independent of PI3K/PDK1 and mTORC1/2 activities (Figure 10). These findings strongly suggest that P2 × 7R may be a fine-tuner of autophagic process in astrocytes by regulating AKT-S473 phosphorylation. Therefore, our findings provide new evidence indicating that P2 × 7R may be considered as a therapeutic approach to modulate astroglial autophagy, which would influence seizure activity and neuroinflammation.

## Figures and Tables

**Figure 1 ijms-21-06476-f001:**
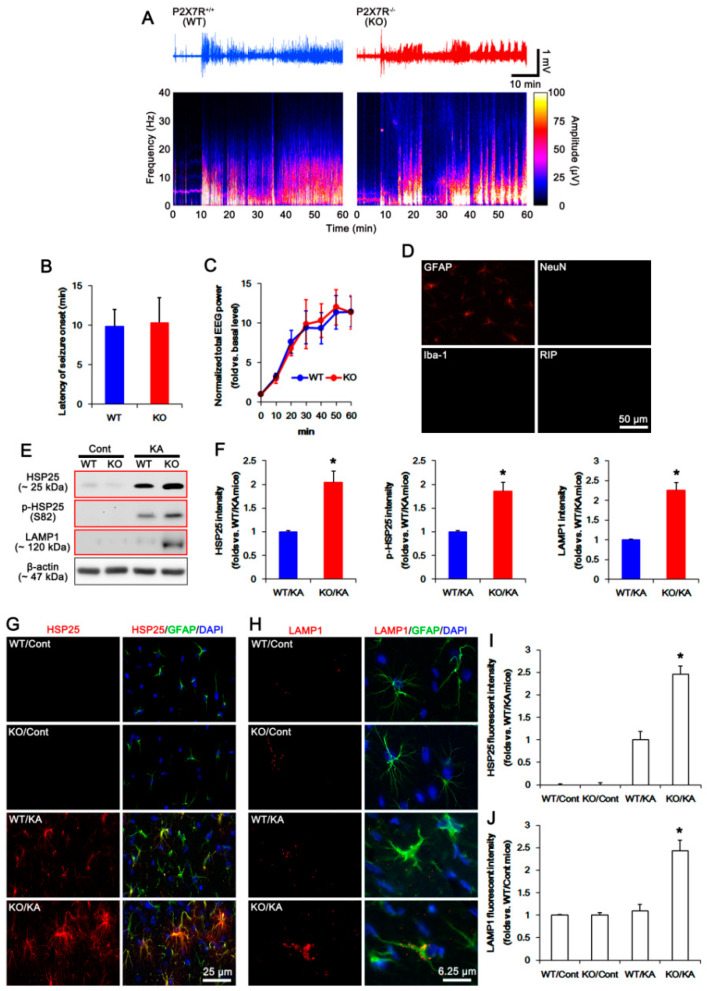
Effects of P2X7 receptor (P2X7R) deletion on seizure activity, heat shock protein 25 (HSP25) expression, and astroglial autophagy in response to kainic acid (KA). (**A**–**C**) Effect of P2X7R deletion on seizure susceptibility in response to KA. P2X7R deletion does not affect the seizure susceptibility in response to KA. (**A**) Representative EEG traces and frequency-power spectral temporal maps. (**B**,**C**) Quantification of the latency of seizure onset (**B**) and total EEG power (**C**) in response to KA. (mean ± SEM; *n* = 7, respectively). (**D**) Identification of isolated cells using glial fibrillary acidic protein (GFAP, an astroglial marker), neuronal nuclear antigen (NeuN, a neuronal marker), ionized calcium binding protein-1 (Iba-1, a microglial marker), and RIP (an oligodendroglial marker) antibodies. (**E**,**F**) Effect of P2X7R deletion on KA-induced HSP25 and LAMP1 expressions in isolated astrocytes. As compared to wild-type (WT) astrocytes, both HSP25 and LAMP1 expression levels are higher in P2X7R knockout (KO) astrocytes 7 days after KA injection (red box). (**E**) Representative Western blot of HSP25 and lysosome-associated membrane protein 1 (LAMP1). Cont, control animals; KA, KA-injected animals. (**F**) Quantifications of HSP25 and LAMP1 expressions and HSP phosphorylation following KA injection (mean ± SEM; ** p* < 0.05 vs. KA-treated WT astrocytes; *n* = 7, respectively). (**G**,**H**) Effect of P2X7R deletion on HSP25 and LAMP1 protein expressions in the hippocampal astrocytes. As compared to WT mice, both HSP25 (**G**) and LAMP1 (**H**) expression levels are higher in P2X7R KO mice 7 days after KA injection. (**I**,**J**) Quantifications of HSP25 (I) and LAMP1 (**J**) fluorescent intensity following KA injection (mean ± SEM; ** p* < 0.05 vs. KA-treated WT mice; *n* = 7, respectively).

**Figure 2 ijms-21-06476-f002:**
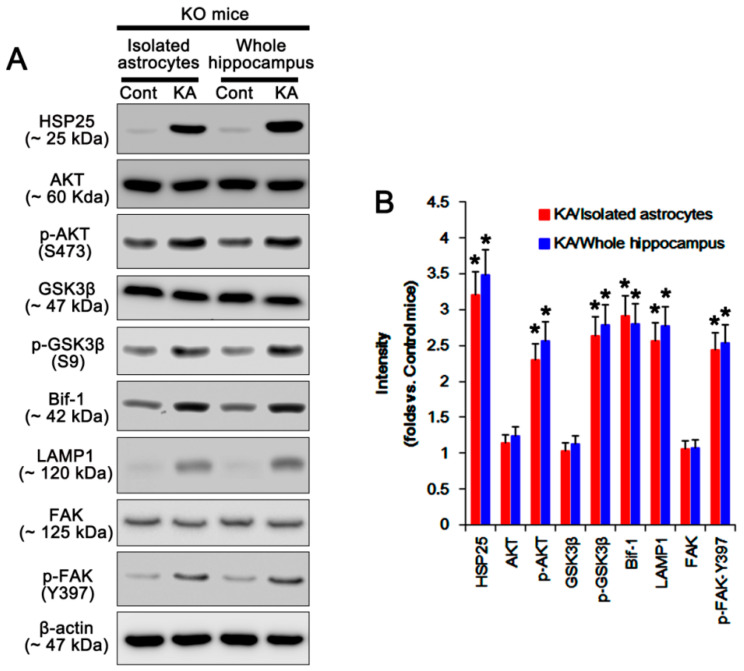
Effects of KA on expressions and phosphorylations of HSP25, AKT (also known as protein kinase B), glycogen synthase kinase-3β (GSK3β), bax interacting factor 1 (Bif-1), LAMP1, and focal adhesion kinase (FAK) in isolated astrocytes and the whole hippocampus obtained from P2X7R mice. (**A**) Representative Western blot of expressions and phosphorylations of HSP25, AKT, GSK3β, Bif-1, LAMP1, and FAK. Cont, control animals; KA, KA-injected animals. (**B**) Quantifications of expressions and phosphorylations of HSP25, AKT, GSK3β, Bif-1, LAMP1, and FAK following KA injection (mean ± SEM; ** p* < 0.05 vs. control mice; *n* = 7, respectively).

**Figure 3 ijms-21-06476-f003:**
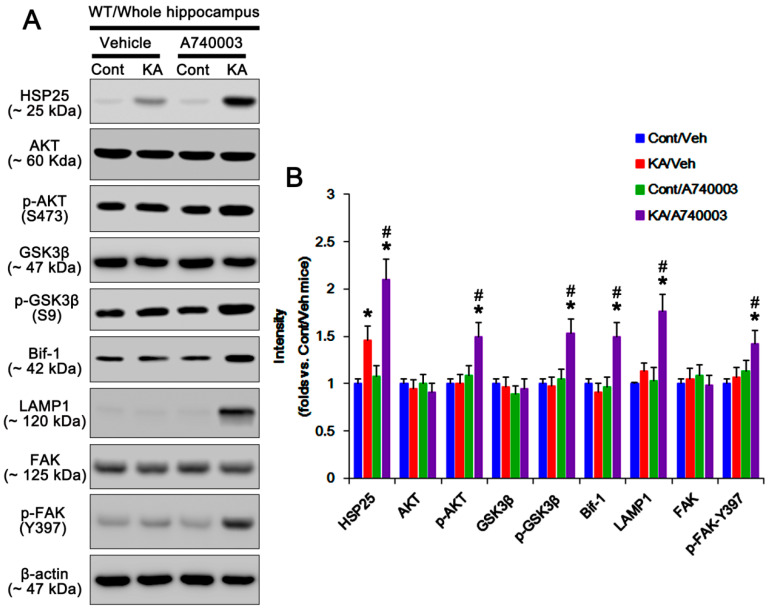
Effects of A740003 on expressions and phosphorylations of HSP25, AKT, GSK3β, Bif-1, LAMP1, and FAK in the whole hippocampus obtained from WT mice following KA injection. (**A**) Representative Western blot of expressions and phosphorylations of HSP25, AKT, GSK3β, Bif-1, LAMP1, and FAK. Cont, control animals; KA, KA-injected animals. (**B**) Quantifications of expressions and phosphorylations of HSP25, AKT, GSK3β, Bif-1, LAMP1, and FAK in KO astrocytes following KA injection (mean ± SEM; **^,#^ p* < 0.05 vs. control and vehicle-treated mice, respectively; *n* = 7, respectively).

**Figure 4 ijms-21-06476-f004:**
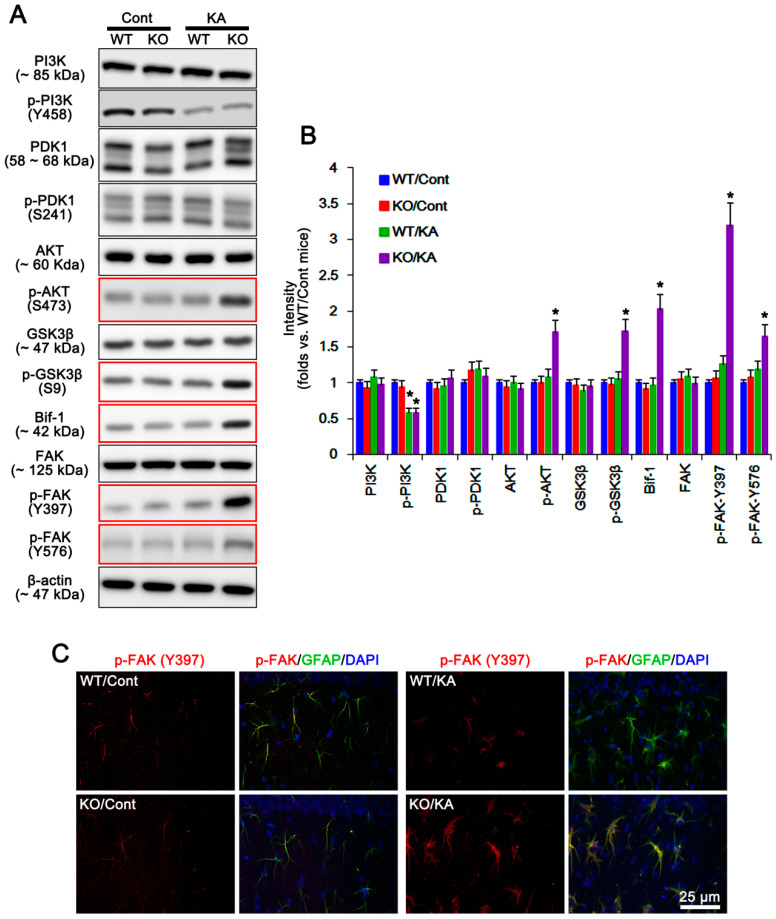
Effects of P2X7R deletion on expressions and phosphorylations of phosphatidylinositol-4,5-bisphosphate 3-kinase (PI3K), phosphoinositide-dependent kinase-1 (PDK1), AKT, GSK3β, Bif-1, and FAK in isolated astrocytes following KA injection. KA increases p-AKT-S473, p-GSK3β-S9, Bif-1, p-FAK-Y397, and p-FAK-Y576 levels in KO astrocytes 7 days after KA injection (red box). (**A**) Representative Western blot of expressions and phosphorylations of PI3K, PDK1, AKT, GSK3β, Bif-1, and FAK. Cont, control animals; KA, KA-injected animals. (**B**) Quantifications of expressions and phosphorylations of PI3K, PDK1, AKT, GSK3β, Bif-1, and FAK following KA injection (mean ± SEM; ** p* < 0.05 vs. control WT astrocytes; *n* = 7, respectively). (**C**) Effect of P2X7R deletion on p-FAK-Y397 level in the hippocampal astrocytes. As compared to WT mice, p-FAK-Y397 level is higher in KO mice 7 days after KA injection.

**Figure 5 ijms-21-06476-f005:**
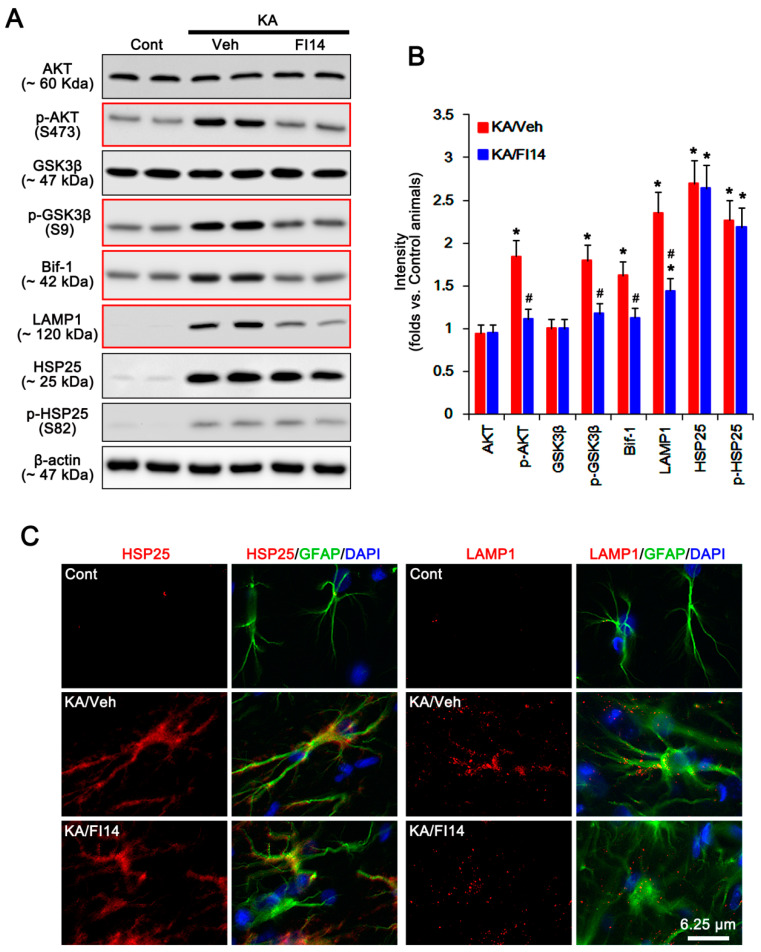
Effects of FAK inhibition on expressions and phosphorylations of AKT, GSK3β, Bif-1, LAMP1, HSP25, and p-HSP25 in KO astrocytes following KA injection. As compared to vehicle, FAK inhibitor 14 (FI14) abolishes the upregulations of p-AKT-S473, p-GSK3β-S9, Bif-1, and LAMP1, but not HSP25, levels 7 days after KA injection (red box). (**A**) Representative Western blot of expressions and phosphorylations of AKT, GSK3β, Bif-1, LAMP1, HSP25, and p-HSP25 in KO astrocytes. Cont, control animals; KA, KA-injected animals; Veh, vehicle; FI4, FAK inhibitor 14. (**B**) Quantifications of expressions and phosphorylations of AKT, GSK3β, Bif-1, LAMP1, HSP25, and p-HSP25 in KO astrocytes following KA injection (mean ± SEM; **^,#^ p* < 0.05 vs. control KO astrocytes and vehicle, respectively; *n* = 7, respectively). (**C**) Effect of P2X7R deletion on HSP25 and LAMP1 expression in the hippocampal astrocytes. As compared to vehicle, FI14 abrogates LAMP1, but not HSP25, expression in KO mice 7 days after KA injection.

**Figure 6 ijms-21-06476-f006:**
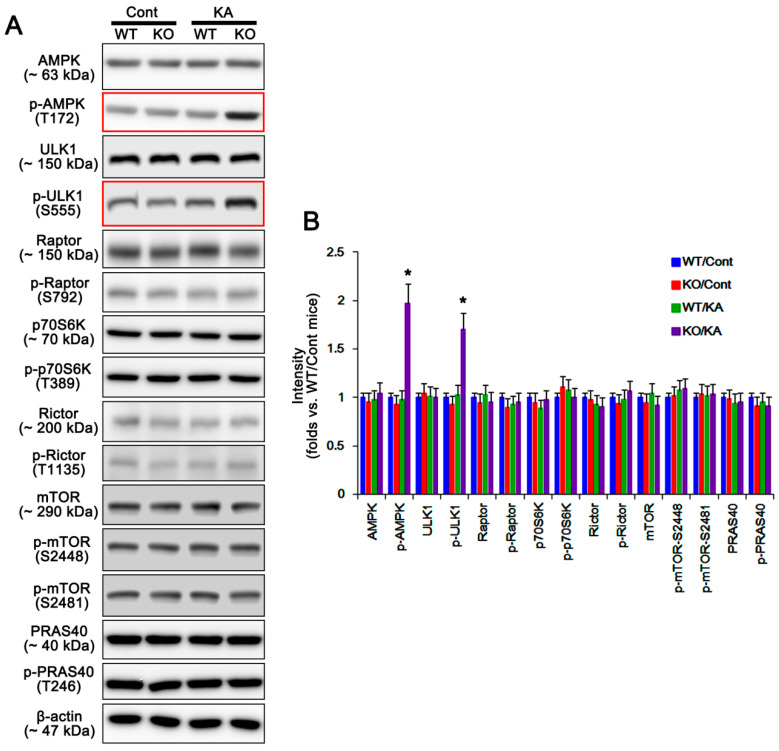
Effects of P2X7R deletion on expressions and phosphorylations of 5′ adenosine monophosphate-activated protein kinase (AMPK), unc-51 like autophagy activating kinase 1 (ULK1), regulatory associated protein of mammalian target of rapamycin (Raptor), p70S6 kinase (p70D6K), rapamycin-insensitive companion of mammalian target of rapamycin (Rictor), mammalian target of rapamycin (mTOR), and proline-rich AKT substrate of 40 kDa (PRAS40) in isolated astrocytes following KA injection. KA increases p-AMPK and p-ULK1 levels in KO astrocytes 7 days after KA injection (red box). (**A**) Representative Western blot of expressions and phosphorylations of AKT, GSK3β, Bif-1, LAMP1, HSP25, and p-HSP25 in KO astrocytes. Cont, control animals; KA, KA-injected animals. (**B**) Quantifications of expressions and phosphorylations of AMPK, ULK1, Raptor, p70D6K, Rictor, mTOR, and PRAS40 in isolated astrocytes following KA injection (mean ± SEM; ** p* < 0.05 vs. control WT astrocytes; *n* = 7, respectively).

**Figure 7 ijms-21-06476-f007:**
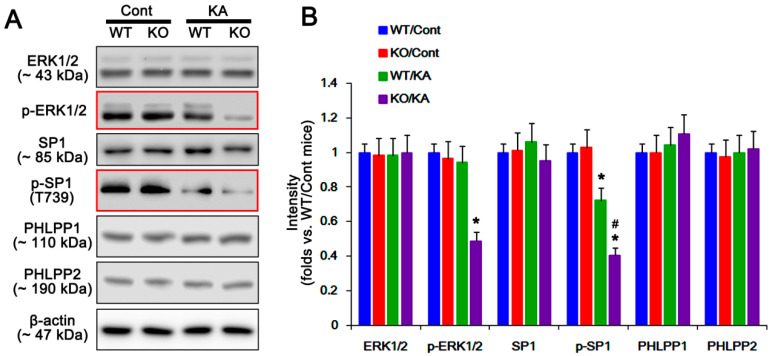
Effects of P2X7R deletion on expressions and phosphorylations of extracellular regulated kinase 1/2 (ERK1/2), specificity protein 1 (SP1), pleckstrin homology domain, and leucine-rich repeat protein phosphatase 1 (PHLPP1) and PHLPP2 in isolated astrocytes following KA injection. As compared to WT astrocytes, KA reduces p-ERK1/2 and p-SP1 levels in KO astrocytes 7 days after KA injection (red box). (**A**) Representative Western blot of expressions and phosphorylations of ERK1/2, SP1, PHLPP1, and PHLPP2 in isolated astrocytes following KA injection. Cont, control animals; KA, KA-injected animals. (**B**) Quantifications of expressions and phosphorylations of ERK1/2, SP1, PHLPP1, and PHLPP2 in isolated astrocytes following KA injection (mean ± SEM; **^,#^ p* < 0.05 vs. Control WT astrocytes and KA-treated WT astrocytes, respectively; *n* = 7, respectively).

**Figure 8 ijms-21-06476-f008:**
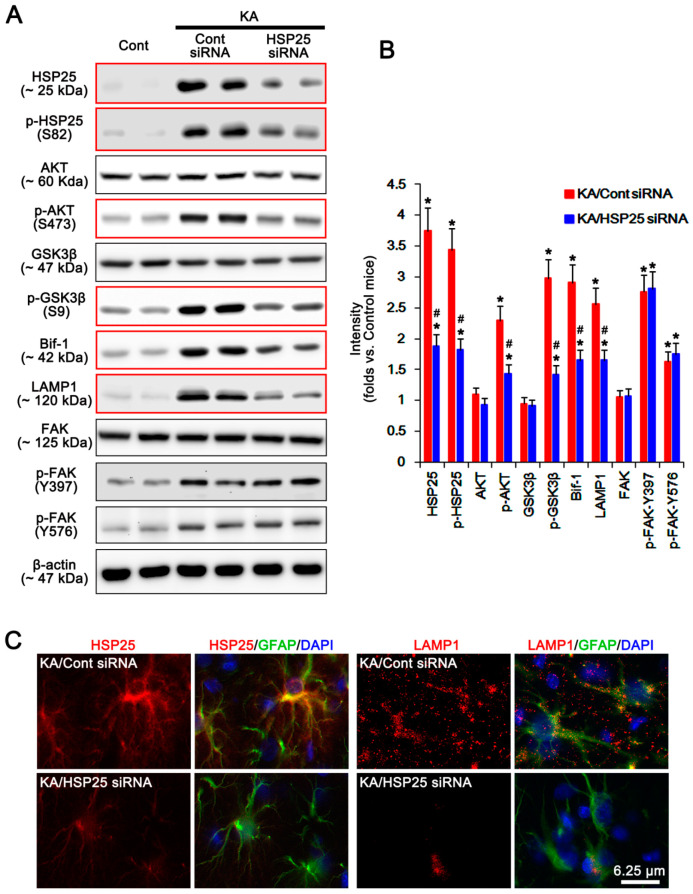
Effects of HSP25 knockdown on expressions and phosphorylations of HSP25, AKT, GSK3β, Bif-1, LAMP1, and FAK in KO astrocytes following KA injection. HSP25 knockdown abrogates upregulations of HSP25, p-HSP25, p-AKT-S473, p-GSK3β-S9, Bif-1, and LAMP1 induced by KA, while it does not affect FAK and its phosphorylations, 7 days after KA injection (red box). (**A**) Representative Western blot of expressions and phosphorylations of HSP25, AKT, GSK3β, Bif-1, LAMP1, and FAK in KO astrocytes. Cont, control animals; KA, KA-injected animals. (**B**) Quantifications of expressions and phosphorylations of HSP25, AKT, GSK3β, Bif-1, LAMP1, and FAK in KO astrocytes following KA injection (mean ± SEM; **^,#^ p* < 0.05 vs. control KO astrocytes and control siRNA, respectively; *n* = 7, respectively). (**C**) Effects of HSP25 knockdown on HSP25 and LAMP1 expression in the hippocampal astrocytes. As compared to control siRNA, FI14 abrogates LAMP1, but not HSP25, expression in KO mice 7 days after KA injection.

**Figure 9 ijms-21-06476-f009:**
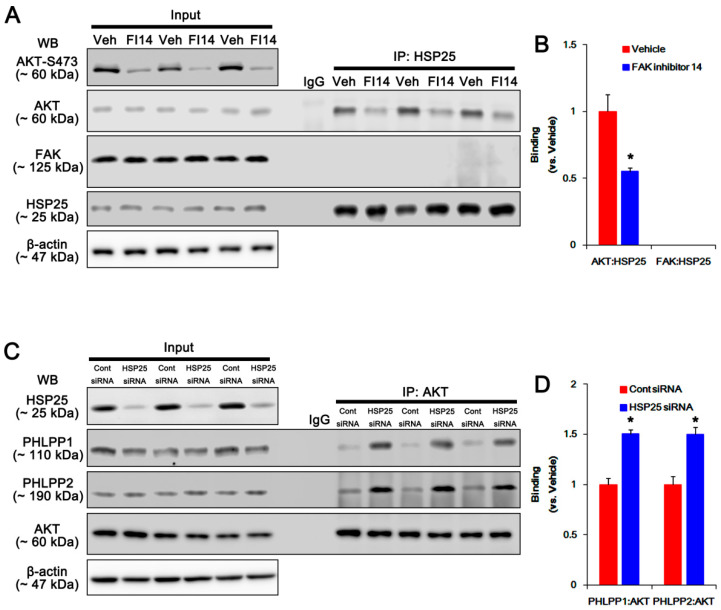
Effects of FAK inhibition and HSP25 knockdown on the bindings of AKT-HSP25, FAK-HSP25, PHLPP1-AKT, and PHLPP1-AKT in the hippocampus of KO mice following KA injection. (**A**,**B**) Effects of FI14 on the bindings of AKT-HSP25 and FAK-HSP25. FI14 attenuates AKT-HSP25 binding with reducing AKT-S473 phosphorylation, 7 days after KA injection. FAK does not bind with HSP25. (**A**) Representative Western blot of p-AKT-S473, AKT, FAK, and HSP25 in the KO hippocampus. Veh, vehicle; FI14, FAK inhibitor 14. (**B**) Quantifications of the bindings of AKT-HSP25 and FAK-HSP25 (mean ± SEM; ** p* < 0.05 vs. vehicle, respectively; *n* = 7, respectively). (**C**,**D**) Effects of HSP25 knockdown on the bindings of PHLPP1-AKT and PHLPP2-AKT. HSP25 siRNA ameliorates PHLPP1-AKT and PHLPP2-AKT bindings, 7 days after KA injection. (**C**) Representative Western blot of HSP25, PHLPP1, PHLPP2, and AKT in the KO hippocampus. (**D**) Quantifications of the bindings of PHLPP1-AKT and PHLPP2-AKT (mean ± SEM; ** p* < 0.05 vs. vehicle, respectively; *n* = 7, respectively).

**Figure 10 ijms-21-06476-f010:**
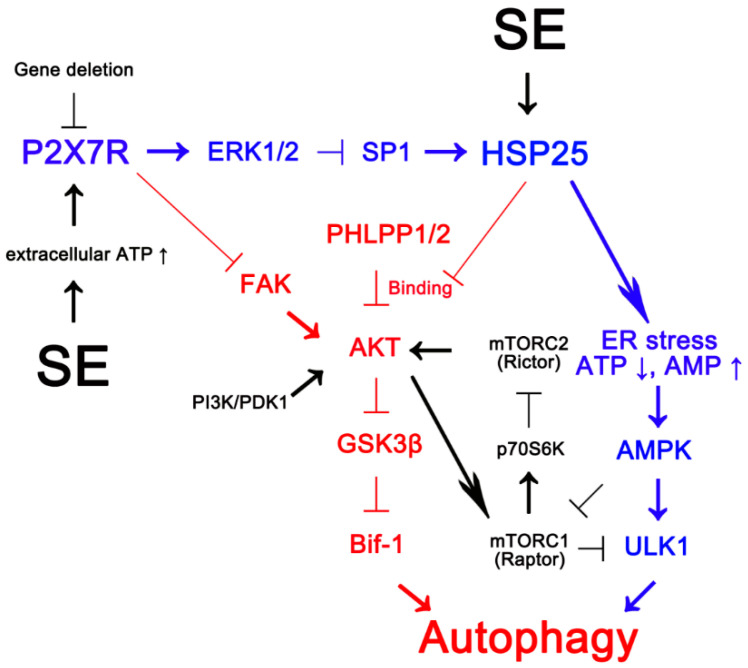
Scheme of inhibitory role of P2X7R in AKT-S473 hyperphosphorylation during astroglial autophagy induced by KA injection based on the present data (red) and previous reports (blue and black) [6,14,21,22,23]. After KA injection, P2X7R activation inhibits FAK phosphorylation and HSP25 transactivation via ERK1/2-mediated SP1-T739 phosphorylation. P2X7R deletion leads to sustained HSP25 expression, which activates AMPK/ULK1-mediated astroglial autophagy (blue) [6]. In addition, P2X7R deletion increases FAK autophosphorylation [20]. Subsequently, the activated FAK phosphorylates AKT, and the prolonged HSP25 expression abrogates the binding of PHLPP1/2 to AKT, which result in AKT-S473 hyperphosphorylation. Sustained AKT-S473 phosphorylation exerts AKT/GSK3β-mediated Bif-1 induction that triggers astroglial autophagy (red), independent of PI3K/PDK1 and mTOR complex (mTORC) 1/2 activities (black). Thus, these findings suggest that P2X7R may be a fine-tuner of autophagic process in astrocytes by regulating AKT-S473 phosphorylation.

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
