# Peer review of "P2 × 7 Receptor Inhibits Astroglial Autophagy via Regulating FAK- and PHLPP1/2-Mediated AKT-S473 Phosphorylation Following Kainic Acid-Induced Seizures"

_ijms, 2020, doi:10.3390/ijms21186476_

Round 1

Reviewer 1 Report

The authors provide a manuscript entitled “P2X7 receptor inhibits astroglial autophagy via regulating FAK- and PHLPP1/2-mediated AKT-S473 phosphorylation”. The authors elucidate a signaling pathway involving absence of P2X7 receptors in kainic acid treated mice and subsequent hyper-phosphorylation of AKT mediated by FAK. The authors use western blot, co-IP and immunohistochemistry assays to substantiate their arguments.

Overall, the topic is interesting, and the findings are conclusive. However, the manuscript was a bit difficult to read and to follow, therefore I have a few suggestions to improve the article.

First of all, it took me until row 81 (and after reading the methods section) to find out that the authors utilize knock-out mice in their experiments. It would be very helpful to the readers if the authors could mention early on that they use P2X7 KO mice (or P2X7-/- mice). The authors frequently write “P2X7 deletion” – I would prefer, if the authors could change this to P2X7-/- or P2X7 KO.

Occasionally, the authors have problems with their citation style: e.g. row 53, 80 or 260.

Results section:

The authors describe Fig 1A-C in only one sentence. Combined with the lack of description in the methods section, it is completely unclear what this experiment is good for and how it was performed. It is never mentioned how the latency was determined. What kind of latency is it? How was the power evaluated and calculated? What do the authors mean by “…after establishing a stable baseline for at least 30 min” (row 384, methods section)? – is that the latency?

Fig 1F: the authors claim that HSP25 expression is more pronounced in KO mice than in WT mice “data not shown” (row 86) – this is unacceptable! The authors must show their data. It is highly suspicious if the authors do not show their data here, but they do so in Fig 1G.

Fig 1E: what is the unit on the Y-axis? If it is normalized, the authors should label the axis accordingly. The authors compare HSP25, p-HSP25 and LAMP-1 vs. WT/KA – it would be good, if the authors showed these bars as well.

Figs 1F and G: is there any form of quantification possible?

Row 112: “These findings confirm that…” – this should rather be mentioned in the discussion section.

Row 122: From the discussion section I gathered that phosphorylation of FAK Y397 is mediated by auto-phosphorylation -it would be worth mentioning here as well or mention the kinase that should lead to Y397 and Y576 phosphorylation.

Fig 2B, 3B, 4B, 5B, 6B, 7B,D: what is the unit on the Y-axis?

Fig 5B: the authors show reduced p-ERK1/2 and p-SP1 intensity in kainic acid treated P2X7 KO mice – however, the authors state that P2X7R deletion does not involve ERK1/2 and SP-1 mediated phosphorylation: can the authors explain why?

Section 2.5 “HSP25 binding to AKT abrogates AKT-S473 dephosphorylation”

The first part of this section describes Fig 6, but the description of Fig 6C is missing.

Fig 8: I think it is a good idea to summarize the findings of this study and previous studies. However, it is still a bit unclear and confusing. What does “SE” stand for? Is the “SE” on the left the same as the “SE” on the right?

If I follow the path: the left “SE” leads to increased extracellular ATP which activates P2X7 receptors. This activates ERK1/2 which inhibits SP1, which in turn does not activate HSP25. This fits the results of this study that deletion of P2X7 increases HSP25. However, I am wondering what the “SE” on the right stands for? Does this induce HSP25 – why would it do that? Then “SE” would cause two opposing effects?

What is the relation of Bif-1 and autophagy?

Discussion section:

The authors mention (row 288) that P2X7 antagonists induce phosphorylation downstream of PI3K. The authors mention that their results are independent of the PI3K pathway. So what happens to the authors pathway if a P2X7 specific inhibitor (e.g. AZ10606120) is used? What happens if a more or less specific agonist like BzATP is used?

Row 319:

Can the authors speculate why mTORC2 associated AKT activity is unaffected in their study?

Row 327:

The authors mention that P2X7 deletion does not affect PHLPP1/2 transactivation. What does the increased binding (Fig 7) mean?

Row 339:

The authors mention that HSP25 silencing prevents autophagy. The authors never show that their model induces autophagy, hence it does not show that it prevents autophagy neither.

I have several suggestions to improve the manuscript:

First, I would like to see how the model induces autophagy and how e.g. silencing HSP25 reduces it.

I would like to see, if the authors can reproduce their results in wt mice in presence of a P2X7 antagonist (e.g. AZ10606120) and if the opposing results can be obtained when the authors apply BzATP.

Then the authors really need to improve the methods section: It is completely unclear how and why the EEG recordings were performed.

It is not clear to me why the authors have a separate co-IP section, but not a separate western blot section.

In their western blot section, the authors mention that they use astrocytes from the whole cortex to get enough material. Why did the authors use the whole hippocampus (including neurons?) for their co-IP experiments – why didn’t they use astrocytes from the entire cortex as in their western blot experiments?

In row 418, the authors mention that they use “control and test animals”… what do they mean by that? – please specify!

What is the manufacturer of the manufacturer of the microscope?

Statistics section: The authors mention that they use student’s t-tests and ANOVAs for their calculations. The authors should specify in which instances they used a t-test and in which they used an ANOVA.

Author Response

Dear Reviewer 1;

I am enclosing herewith the revised version of our manuscript entitled P2X7 receptor inhibits astroglial autophagy via regulating FAK- and PHLPP1/2-mediated AKT-S473 phosphorylation (Ms. No.: ijms-891533).

We appreciate the reviewer’s recommendations to improve our manuscript.

With respect to reviewers’ comments and suggestions, we performed additional experiments and addressed the reviewers' points. Furthermore, we carefully re-edited text bases on reviewers’ comments. My responses to comments are as followed;

Reviewer 1

  1. First of all, it took me until row 81 (and after reading the methods section) to find out that the authors utilize knock-out mice in their experiments. It would be very helpful to the readers if the authors could mention early on that they use P2X7 KO mice (or P2X7-/- mice). The authors frequently write “P2X7 deletion” – I would prefer, if the authors could change this to P2X7-/- or P2X7 KO.

With respect to reviewer’s comments, we have corrected them.

  1. Occasionally, the authors have problems with their citation style: e.g. row 53, 80 or 260.

With respect to reviewer’s comments, we have corrected them.

  1. The authors describe Fig 1A-C in only one sentence. Combined with the lack of description in the methods section, it is completely unclear what this experiment is good for and how it was performed. It is never mentioned how the latency was determined. What kind of latency is it? How was the power evaluated and calculated? What do the authors mean by “…after establishing a stable baseline for at least 30 min” (row 384, methods section)? – is that the latency?

With respect to reviewer’s comments, we have rewritten these sentences as followed;

“The degree of seizure activity affects astroglial responses including HSP25 induction [16]. Thus, we evaluated the seizure susceptibilities of wild-type (WT) and P2X7R KO mice in responses to KA. Consistent with our previous report [6], there was no difference in the latency of seizure onset and the normalized total power of electroencephalogram (EEG) during seizures in response to KA between WT and P2X7R KO mice (Fig. 1A-C). .”

In addition, we have described how to evaluate and latency of seizure onset and total EEG power in the materials and methods sections as followed;

“Three days after surgery, mice were given a single dose of KA aforementioned after establishing a stable baseline for at least 30 minutes, and latency or seizure onset and total power were measured from each animal. EEG signals were recorded with a DAM 80 differential amplifier (0.1–1000 Hz bandpass; World Precision Instruments, Sarasota, FL, USA) and the data were digitized (1000 Hz) and analyzed using LabChart Pro v7 software (ADInstruments, Bella Vista, New South Wales, Australia). Latency of seizure onset was defined as the time point showing more than 3 s and consisting of a rhythmic discharge between 4 and 10 Hz with amplitude of at least two times higher than the baseline EEG. Total EEG power was normalized by the baseline power obtained from each animal. Spectrograms were automatically calculated using a Hanning sliding window with 50% overlap by LabChart Pro v7 [6,64].”

  1. Fig 1F: the authors claim that HSP25 expression is more pronounced in KO mice than in WT mice “data not shown” (row 86) – this is unacceptable! The authors must show their data. It is highly suspicious if the authors do not show their data here, but they do so in Fig 1G.

→ We had described that P2X7R was rarely observed in both animals under physiological condition (data not shown), but not KA-treated condition (provided in a previous version). With respect to reviewer’s comments, however, we have provided photos concerning astroglial HSP25 expression as Fig. 1G.

  1. Fig 1E: what is the unit on the Y-axis? If it is normalized, the authors should label the axis accordingly. The authors compare HSP25, p-HSP25 and LAMP-1 vs. WT/KA – it would be good, if the authors showed these bars as well.

With respect to reviewer’s comments, we have changed graphs.

  1. Figs 1F and G: is there any form of quantification possible?

With respect to reviewer’s comments, we have inserted the quantification data concerning HSP25 fluorescent intensity.

  1. Row 112: “These findings confirm that…” – this should rather be mentioned in the discussion section.

With respect to reviewer’s comments, we have changed this sentence and discussed in the text.

  1. Row 122: From the discussion section I gathered that phosphorylation of FAK Y397 is mediated by auto-phosphorylation -it would be worth mentioning here as well or mention the kinase that should lead to Y397 and Y576 phosphorylation.

With respect to reviewer’s comments, we have changed these sentence as followed;

 “On the other hand, FAK is one of the kinases for AKT, which is required for autophagy [18]. FAK activation is required for the phosphorylations on several tyrosine (Y) residues in kinase domain, including Y397, 576 and 577. Among them, Y397 is autophosphorylation site, and other sites are phosphorylated by Src [19].”

  1. Fig 2B, 3B, 4B, 5B, 6B, 7B,D: what is the unit on the Y-axis?

With respect to reviewer’s comments, we have inserted the unit (folds vs. WT/Cont mice, Control mice or vehicle.

  1. Fig 5B: the authors show reduced p-ERK1/2 and p-SP1 intensity in kainic acid treated P2X7 KO mice – however, the authors state that P2X7R deletion does not involve ERK1/2 and SP-1 mediated phosphorylation: can the authors explain why?

→ In our previous study [6], P2X7R deletion promotes astroglial autophagy by prolonged HSP25 transactivation due to abrogation of ERK1/2-mediated SP1-T739 phosphorylation. This is because HSP25 expression relies on SP1 binding to its promoter region, and SP1-T739 phosphorylation reduces the SP1 DNA-binding ability [26,27]. SP1 also increases PHLPP promoter activity and its protein level, and facilitates AKT dephosphorylation [28]. Therefore, it is likely that P2X7R would regulate AKT-S473 phosphorylation by inhibiting ERK1/2-SP1-mediated PHLPP1/2 transactivations following KA injection. In the present study, however, KA injection did not affect expression levels of PHLPP1 and PHLPP2 in both WT and KO astrocytes, although KA decreases ERK1/2 and SP1-T739 phosphorylations in the P2X7R KO astrocytes more than in WT astrocytes. These findings suggest that P2X7R deletion may not affect ERK1/2-SP1-mediated PHLPP1/2 transactivations, although P2X7R knockout facilitates ERK1/2-SP1-medaited HSP25 overexpression. We have described this in the text.

  1. Section 2.5 “HSP25 binding to AKT abrogates AKT-S473 dephosphorylation” The first part of this section describes Fig 6, but the description of Fig 6C is missing.

With respect to reviewer’s comments, we have inserted the description of Fig. 6C (Fig. 8C in this revised version) in the text.

  1. Fig 8: I think it is a good idea to summarize the findings of this study and previous studies. However, it is still a bit unclear and confusing. What does “SE” stand for? Is the “SE” on the left the same as the “SE” on the right? If I follow the path: the left “SE” leads to increased extracellular ATP which activates P2X7 receptors. This activates ERK1/2 which inhibits SP1, which in turn does not activate HSP25. This fits the results of this study that deletion of P2X7 increases HSP25. However, I am wondering what the “SE” on the right stands for? Does this induce HSP25 – why would it do that? Then “SE” would cause two opposing effects?

→ Since we believe that ERK1/2-SP1 is not unique signaling pathway for HSP25 induction, we had inserted SE on the right side. With respect to reviewer’s comments, it is good for the description of the results in the present study in Fig. 8 (Fig. 10 in this revised version). Thus, we have changed SE to seizures and corrected the Fig. 8 (Fig. 10 in this revised version).

  1. What is the relation of Bif-1 and autophagy?

→ Bif-1 plays an important role in GSK3β-mediated autophagic process [13], since Bif-1 modulates autophagy by regulating autophagosome formation [38]. Considering AKT-mediated GSK3β phosphorylation (reducing its activity) [39], it is likely that prolonged AKT activation may lead to Bif-1-mediated astroglial autophagy after KA injection. Indeed, KA increased GSK3β phosphorylation and Bif-1 expression in P2X7R KO astrocytes, accompanied by enhanced LAMP1 expression. With respect to reviewer’s comments, we have described this in the text as followed;

“In the present study, P2X7R KO astrocytes showed the increases in GSK3β phosphorylation and expressions of Bif-1 and LAMP1, accompanied by AKT-S473 hyper-phosphorylation after KA treatment. Since AKT-mediated GSK3β phosphorylation (reducing its activity) transactivates Bif-1 that modulates autophagy by regulating autophagosome formation [13,38,39], our findings indicate that AKT-S473 hyper-phosphorylation may promote GSK3β/Bif-1-mediated autophagic pathway in P2X7R KO astrocytes after KA injection.”

  1. The authors mention (row 288) that P2X7 antagonists induce phosphorylation downstream of PI3K. The authors mention that their results are independent of the PI3K pathway. So what happens to the authors pathway if a P2X7 specific inhibitor (e.g. AZ10606120) is used? What happens if a more or less specific agonist like BzATP is used?

→ In our previous study [6], BzATP (a P2X7R agonist) reduced HSP25 induction, while P2X7R antagonists (oxATP and A740003) elevated HSP25 and LAMP1 expressions after KA injection. Furthermore, BzATP induced astroglial apoptosis after KA injection, while oxATP and A740003 did not. These findings suggest that P2X7R inhibition may accelerate HSP25 induction and autophagy, while P2X7R activation may induce apoptosis in astrocytes accompanied by reduction in HSP25 expression. In this revised version, we have performed additional experiments applying A740003, but not BzATP (due to its pro-apoptotic effects), to WT mice. Similar to the case of P2X7R mice, A740003 facilitated AKT/GSK3β/Bif-1 autophagic pathway in WT mice. We have inserted these data in the text as Fig. 3 in this revised version.

In the canonical pathway, AKT activation requires PI3K-dependent phosphorylation of threonine (T) 308 and S473 sites. When phosphatidylinositols are produced by a PI3K, PDK1 is recruited to the plasma membrane and leads to AKT-T308 phosphorylation [33]. The AKT-S473 phosphorylation is regulated by PDK1 or mTORC2, once known as the elusive PDK2 [33]. Consistent with our previous study [6], the present study demonstrates that KA specifically increased astroglial AKT-S473 phosphorylation in P2X7R KO astrocytes. However, KA did not affect PDK1-S241 phosphorylation in both WT and P2X7R KO astrocytes, although it reduced PI3K activity (phosphorylation) in both groups. Since PDK1-S241 site is constitutively phosphorylated by an autophosphorylation reaction in trans and PDK1 does not always require phosphatidylinositols for its activities [34,35], our findings indicate that the PI3K/PDK1 pathway may not be involved in KA-induced AKT-S473 hyper-phosphorylation in P2X7R KO astrocytes. However, mitogen-activated protein kinase 2 (MK2) and a complex with the integrin-linked kinase (ILK) and interact with protein kinase C-ζ (PKC-ζ) can phosphorylate AKT on S473 site independent of PI3K/PDK1 [31,36,37]. Thus, it is not excluded the possibility that these non-canonical signaling pathways would be involved in AKT-S473 hyper-phosphorylation in P2X7R KO astrocytes.

On the other hand, P2X7R inhibition increases FAK activity by the autophosphorylation of FAK at Y397 site [20]. FAK is also responsible for AKT-S473 phosphorylation via PI3K/PDK1-independent signaling opapathway [43,44]. In the present study, KA up-regulated FAK-Y397 and -Y576 phosphorylations in P2X7R KO astrocytes. KA elevated Y397 phosphorylation more than Y576 phosphorylation. Furthermore, FI14 inhibited autophagy in P2X7R KO astrocytes with diminishing phosphorylations of AKT-S473 and GSK3β-S9, and expressions of Bif-1 and LAMP1 following KA injection. Since FAK directly binds to AKT and phosphorylates S473 site independent of PI3K/PDK1 activity [43,44], our findings indicate that P2X7R deletion may increase FAK autophosphorylation in response to KA, which is required for AKT-mediated astroglial autophagy, independent of PI3K/PDK1 signaling pathway. We have discussed these in the text.

  1. Row 319: Can the authors speculate why mTORC2 associated AKT activity is unaffected in their study?

→ In the present study, KA increased AMPK activity (T172 phosphorylation) in P2X7R KO astrocytes. However, KA did not affect phosphorylations of Raptor, Rictor, PRAS40, p70S6K and mTOR in P2X7R KO astrocytes. Thus, our findings suggest that P2X7R deletion may facilitate astroglial autophagy independent of mTORC1/2 pathways, and that mTORC2 may not participate in AKT-S473 hyper-phosphorylation in P2X7R KO astrocytes following KA injection. However, Rictor can form a complex with the ILK and interact with PKC-ζ, which regulate AKT-S473 phosphorylation in an mTORC2-independent fashion [36,37]. Thus, it is presumable that ILK- and/or PKC-ζ-mediated signaling pathway may be involved in AKT hyper-activation in P2X7R KO astrocytes following KA injection. In addition, FAK functionally suppresses mTORC2, but not mTORC1 [46]. FAK is also required for optimal signaling to the PDK1/AKT/p70S6K pathway independent of mTOR during autophagy process [47]. In the present study, we found that KA could not affect mTOR, Raptor and Rictor phosphorylations, but increased FAK activity in P2X7R KO astrocytes. Thus, our findings also suggest that increased FAK activity in P2X7R KO astrocytes may lead to KA-induced AKT-S473 hyper-phosphorylation, independent of mTORC1 and 2 activities. We have discussed these in the text.

  1. Row 327: The authors mention that P2X7 deletion does not affect PHLPP1/2 transactivation. What does the increased binding (Fig 7) mean?

→ HSP25 functions as a molecular chaperone [26], and forms complexes with AKT [30]. In addition, p-HSP25 is responsible for a new complex with AKT securing AKT’s natural conformation and enzymatic activity [30]. Indeed, the amounts of HSP25-AKT binding correlates with AKT protein kinase activity [29], and HSP25 siRNA inhibits AKT phosphorylation [6,48]. However, HSP25-AKT interaction is not required to promote AKT activation [49], although HSP25 regulates AKT-S473 phosphorylation [31]. Furthermore, HSP25 does not interact with PI3K at base line or after metabolic stress even in cells with increased HSP25 expression [49]. In the present study, P2X7R deletion and A740003 resulted in prolonged astroglial HSP25 expression following KA injection, which was concomitant with increased HSP25 phosphorylation. Consistent with our previous study [6], silencing HSP25 abolished AKT-S473 and GSK3β-S9 phosphorylations, and diminished Bif-1 and LAMP1 expressions in P2X7R KO mice following KA injection. Furthermore, HSP25 siRNA increased AKT-PHLPP1/2 bindings without altering their expression levels. Considering that HSP25 acts as a scaffolding protein to AKT and MK2 in a manner that requires HSP25-AKT interaction [31], our findings suggest that HSP25 may play a role as a chaperon to abolish AKT-PHLPP1/2 bindings rather than an indispensable factor for AKT phosphorylation, which maintains AKT-S473 phosphorylation in biologically active conformation by inhibiting PHLPP1/2-mediated dephosphorylation. We have described these in the text.

  1. Row 339: The authors mention that HSP25 silencing prevents autophagy. The authors never show that their model induces autophagy, hence it does not show that it prevents autophagy neither. I have several suggestions to improve the manuscript: First, I would like to see how the model induces autophagy and how e.g. silencing HSP25 reduces it.

We have showed the effect of HSP25 silencing as Fig. 8 in this revised version.

  1. I would like to see, if the authors can reproduce their results in wt mice in presence of a P2X7 antagonist (e.g. AZ10606120) and if the opposing results can be obtained when the authors apply BzATP.

With respect to reviewer’s comments, we have performed additional experiments applying A740003, but not Bz ATP (due to its pro-apoptotic effects), to WT mice. Similar to the case of P2X7R mice, A740003 facilitated AKT/GSK3β/Bif-1 autophagic pathway in WT mice. We have inserted these data in the text as Fig. 3.

  1. Then the authors really need to improve the methods section: It is completely unclear how and why the EEG recordings were performed.

With respect to reviewer’s comments, we have rewritten the methods concerning EEG recording as followed;

“Three days after surgery, mice were given a single dose of KA aforementioned after establishing a stable baseline for at least 30 minutes, and latency or seizure onset and total power were measured from each animal. EEG signals were recorded with a DAM 80 differential amplifier (0.1–1000 Hz bandpass; World Precision Instruments, Sarasota, FL, USA) and the data were digitized (1000 Hz) and analyzed using LabChart Pro v7 software (ADInstruments, Bella Vista, New South Wales, Australia). Latency of seizure onset was defined as the time point showing more than 3 s and consisting of a rhythmic discharge between 4 and 10 Hz with amplitude of at least two times higher than the baseline EEG. Total EEG power was normalized by the baseline power obtained from each animal. Spectrograms were automatically calculated using a Hanning sliding window with 50% overlap by LabChart Pro v7 [6,64].”

  1. It is not clear to me why the authors have a separate co-IP section, but not a separate western blot section.

With respect to reviewer’s comments, we have rewritten the methods concerning Western blot and co-IP section.

  1. In their western blot section, the authors mention that they use astrocytes from the whole cortex to get enough material. Why did the authors use the whole hippocampus (including neurons?) for their co-IP experiments – why didn’t they use astrocytes from the entire cortex as in their western blot experiments?

→ As we had mentioned, the amount of isolated astrocytes was insufficient to perform co-immunoprecipitation. Thus, we used the whole hippocampus for co-immunoprecipitation. In this revised version, we have also used the whole hippocampus to address the possibility of the distinct astroglial responses between the cortical and the hippocampal astrocytes. Consistent with our previous study [6] and the immunohistochemical data in the present study (Fig. 1G-J), the Western blot data obtained from whole hippocampus were similar to those from isolated astrocytes (as Fig. 2 in this revised version). Thus, we believe that the data concerning isolated astrocytes may represent the responses of hippocampal astrocytes. We have described these in the text.

  1. In row 418, the authors mention that they use “control and test animals”… what do they mean by that? – please specify!

With respect to reviewer’s comments, we have corrected this as “control and KA-treated animals.

  1. What is the manufacturer of the manufacturer of the microscope?

→ With respect to reviewer’s comments, we have inserted the manufacturer in the text.

  1. Statistics section: The authors mention that they use student’s t-tests and ANOVAs for their calculations. The authors should specify in which instances they used a t-test and in which they used an ANOVA.

With respect to reviewer’s comments, we have changed this section as followed;

“After evaluating the values on normality using Shapiro–Wilk W-test, Student's t-test (comparisons of latency of seizure onset), repeated measured ANOVA (comparisons of total EEG power) or one-way ANOVA (comparisons of Western blots) were used to analyze statistical significance. Bonferroni’s test was used for post-hoc comparisons. A p-value less than 0.05 was considered to be significant [6].”

In addition, we have indicated the statistical test in the text.

I will be grateful if the manuscript could be reviewed and considered for publication.

Very sincerely yours,

Tae-Cheon Kang, DVM, Ph.D

Department of Anatomy & Neurobiology

College of Medicine

Hallym University

Chunchon 24252

South Korea

Reviewer 2 Report

In this manuscript (ijms-891533), the authors demonstrate that P2X7 receptor (P2X7R) and prolonged heat shock protein 25 (HSP25) expression regulate astrocytic autophagy through Akt pathway in the mouse seizured brain induced by kainic acid injection. They examined autophagy signaling pathway using quite number of antibodies, and finally obtained meaningful data on the novel autophagy signaling pathway. The authors properly designed and well organized the experiments to reach the aim of the study. However, it still seems that there are a few points to be addressed or mentioned as follows. 

1. The title

The data of this study were obtained from mice under the certain condition (the seizure-induced hippocampus by kainic acid), not from mice under normal condition. The present title can be misunderstood as general characteristics of astrocytes everywhere under normal condition in the brain. Therefore, it is strongly recommended that the certain condition be added in the title. 

2. Usage of astrocytes isolated from the cortices 

1) The authors used lysates of astrocytes isolated from the cortices for western blotting. Of course, they mentioned that the reason why they used cortical astrocytes was that the amount of the hippocampal astrocytes was not enough. Under seizure, however, the responses between cortical astrocytes and hippocampal astrocytes are unlikely to be similar. The authors should address this issue.  

2) The authors isolated astrocytes using ACSA-2 microbeads of commercial isolation kit. It would take considerable hours to isolate pure astrocytes, and ACSA-2 antibody bound to the specific surface or transmembrane protein of astrocytes during isolation procedure. The authors need to mention the limitation of this study according to the possible effect on signaling pathway by isolation procedure.  

3) Evaluation about purity of isolated astrocytes is needed using astrocyte-specific markers, since the experiments using isolated astrocytes were main. 

3. Clinical significance

HSP25, a mouse small HSP is known as a counterpart of human HSP27. The authors need to mention the relationship between mouse HSP25 and human HSP27, and the possible role of HSP27 on the astrocyte autophagy in view of clinical significance.  

4. Inconsistency 

The results of Fig 2A and B show that compared with the control, p-AKT, pGSK3b and Bif-1 expressions in astrocytes of P2X7R-/- mice were increased after KA injection. However, the scheme in Fig 8 does not reflect the results of Fig 2A and B; especially, inhibitory signals between AKT, GSK3b and Bif-1 in Fig 8. The authors need to properly revise a part of the scheme. 

Author Response

Dear Reviewer 2;

I am enclosing herewith the revised version of our manuscript entitled P2X7 receptor inhibits astroglial autophagy via regulating FAK- and PHLPP1/2-mediated AKT-S473 phosphorylation (Ms. No.: ijms-891533).

We appreciate the reviewer’s recommendations to improve our manuscript.

With respect to reviewers’ comments and suggestions, we performed additional experiments and addressed the reviewers' points. Furthermore, we carefully re-edited text bases on reviewers’ comments. My responses to comments are as followed;

Reviewer 2

  1. The title

The data of this study were obtained from mice under the certain condition (the seizure-induced hippocampus by kainic acid), not from mice under normal condition. The present title can be misunderstood as general characteristics of astrocytes everywhere under normal condition in the brain. Therefore, it is strongly recommended that the certain condition be added in the title. 

With respect to reviewer’s comments, we have changed the title to “P2X7 receptor inhibits astroglial autophagy via regulating FAK- and PHLPP1/2-mediated AKT-S473 phosphorylation following kainic acid-induced seizures”.

  1. Usage of astrocytes isolated from the cortices 

1) The authors used lysates of astrocytes isolated from the cortices for western blotting. Of course, they mentioned that the reason why they used cortical astrocytes was that the amount of the hippocampal astrocytes was not enough. Under seizure, however, the responses between cortical astrocytes and hippocampal astrocytes are unlikely to be similar. The authors should address this issue.  

With respect to reviewer’s comments, we have performed the additional experiment using whole hippocampus to address the possibility of the distinct astroglial responses between the cortical and the hippocampal astrocytes. Consistent with our previous study [6] and the immunohistochemical data in the present study (Fig. 1G-J), the Western blot data obtained from the whole hippocampus were compatible with those from isolated astrocytes. In addition, A740003 (a specific P2X7R antagonist) also facilitated AKT/GSK3β/Bif-1 autophagic pathway in the whole hippocampus of WT mice, similar to the isolated astrocytes from P2X7R mice. Thus, we believe that the data concerning isolated astrocytes may represent the responses of hippocampal astrocytes. We have inserted these data as Fig. 2 and 3 in this revised version.

2) The authors isolated astrocytes using ACSA-2 microbeads of commercial isolation kit. It would take considerable hours to isolate pure astrocytes, and ACSA-2 antibody bound to the specific surface or transmembrane protein of astrocytes during isolation procedure. The authors need to mention the limitation of this study according to the possible effect on signaling pathway by isolation procedure.  

→ With respect to reviewer’s comments, we have mentioned the limitation of this study according to the possible effect on signaling pathway by isolation procedure as followed;

As the reviewer’s comments, The limitations of the present study are the possibility that the dissociation procedure would affect signaling pathway in astrocytes, and that KA would result in the loss of ATPase Na+/K+ transporting subunit beta 2 (ATP1B2) on the cell body and processes of astrocytes recognized by astrocyte specific cell surface antigen 2 (ACSA-2) antibody [63]. In the present study and our previous study [6], the data obtained from the whole hippocampus were compatible with those from isolated astrocytes. Furthermore, the isolation method using ACSA-2 antibody is regarded as less damaging and more effective with brain tissue, and is highly efficient, yielding 300,000 ultrapure astrocytes per mouse cortex dissociated, with an average viability of 86% and purity of > 95%. In addition, ACSA-2 signal remains constant in the sample obtained from models of a stab wound injury, Alzheimer’s disease, spinal cord injury, stroke and lipopolysaccharide-induced inflammation [63]. Thus, it is likely that immunoaffinity-based method using ACSA-2 antibody may dissociate astrocyte with less affecting signaling pathways and yields of isolation following KA treatment. We described these in the discussion section.

3) Evaluation about purity of isolated astrocytes is needed using astrocyte-specific markers, since the experiments using isolated astrocytes were main. 

With respect to reviewer’s comments, we have inserted the data concerning purity of isolated astrocytes using markers for astrocytes (GFAP), neurons (NeuN), microglia (Iba-1) and oligodendrocytes (RIP) as Fig. 1D.

  1. Clinical significance

HSP25, a mouse small HSP is known as a counterpart of human HSP27. The authors need to mention the relationship between mouse HSP25 and human HSP27, and the possible role of HSP27 on the astrocyte autophagy in view of clinical significance.  

→ With respect to reviewer’s comments, we have mentioned the relationship between mouse HSP25 and human HSP27, and the possible role of HSP27 on the astrocyte autophagy in view of clinical significance as followed;  

“Murine/rodent HSP25 is a homologue of human HSP27. HSP25 induction is an indicative of the early astroglial responses including energy-consuming protein synthesis and is one of the highly sensitive and reliable molecules of full development of status epilepticus (SE). Since HSP25 is the main chaperone in astrocytes and plays a role in blockade of mitochondrial damage and apoptosis in naïve astrocytes, HSP25 induction provides insight into the astroglial invulnerability in response to harmful insults [16,52,53]. Indeed, HSP27 over-expression is a negative prognostic marker of astroglioma, which is regarded as a resistant mechanism for astroglioma cells against anti-tumor agents by activating autophagy [54,55]. In addition, HSP27 activates the autophagy-lysosomal pathway in abnormal astrocytes containing Rosenthal fibers in patients of Alexander disease that is an untreatable and fatal neurodegenerative disorder showing seizures caused by heterozygous mutations of GFAP gene [56,57,58]. Although autophagy is a catabolic and survival process for bulk degradation of aberrant organelles and protein aggregates, excessive or unquenched autophagy results in non-apoptotic programmed cell death (type II programmed cell death) independent of caspase activity in various cells including cancer cells [59,60,61]. Indeed, impaired clearance of HSP25 reduces cell viability in astrocytes [62]. Thus, the proper regulations of HSP25/27 expression and its related astroglial autophagy may be one of potential therapeutic strategies for treatment and/or mitigation of symptoms in patients with Alexander disease, astroglioma and other neurodegenerative disorders with astrocyte involvement.”

  1. Inconsistency 

The results of Fig 2A and B show that compared with the control, p-AKT, pGSK3b and Bif-1 expressions in astrocytes of P2X7R-/- mice were increased after KA injection. However, the scheme in Fig 8 does not reflect the results of Fig 2A and B; especially, inhibitory signals between AKT, GSK3b and Bif-1 in Fig 8. The authors need to properly revise a part of the scheme. 

→ AKT-mediated GSK3β phosphorylation decreases GSK3β activity, which subsequently relieves GSK3β-mediated Bif-1 inhibition. Thus, the inhibitory signals in Fig 8 is correct.

I will be grateful if the manuscript could be reviewed and considered for publication.

Very sincerely yours,

Tae-Cheon Kang, DVM, Ph.D

Department of Anatomy & Neurobiology

College of Medicine

Hallym University

Chunchon 24252

South Korea

Round 2

Reviewer 1 Report

The authors present a revised version of their manuscript entitled "P2X7 receptor inhibits astroglial autophagy via regulating FAK- and PHLPP1/2-mediated AKT-S473 phosphorylation following kainic acid-induced seizures". 

The authors have answered most of the points I have previously raised. I have only a few minor comments:

The results are presented in the following order: reference to fig 2a-b in row 129; reference to fig 3a-b in row 142; reference to fig 4a-b in row 160; afterwards (in row 167!) again reference to fig2 (and in row 169 reference to fig3) - this is a little bit confusing; the authors might want to reconsider the order in which they present their results.

In the figure legend to fig. 10 the authors mention application of kainic acid and the effects of its application. - However, as far as I recall, the authors never explain why the use KA (in order to induce seizures) - maybe the authors want to include this in the introduction. 

I think it would be of benefit to a reader, if the authors could include a list of abbreviations.